# The GHSZ Argument: A Gedankenexperiment Requiring More Denken

**DOI:** 10.3390/e22070759

**Published:** 2020-07-10

**Authors:** Frank Lad

**Affiliations:** University of Canterbury, Department of Mathematics and Statistics, Christchurch 8140, New Zealand; frank.lad@canterbury.ac.nz

**Keywords:** Bell’s inequality, entanglement, local realism, supplementary variables, quantum symmetries

## Abstract

I reassess the gedankenexperiment of Greenberger, Horne, Shimony, and Zeilinger after twenty-five years, finding their influential claim to the discovery of an inconsistency inherent in high dimensional formulations of local realism to arise from a fundamental error of logic. They manage this by presuming contradictory premises: that a specific linear combination of four angles involved in their proposed parallel experiments on two pairs of electrons equals both π and 0 at the same time. Ignoring this while presuming the contradictory implications of these two conditions, they introduce the contradiction themselves. The notation they use in their “derivation” is not sufficiently ornate to represent the entanglement in the double electron spin pair problem they design, confounding their error. The situation they propose actually motivates only an understanding of the full array of symmetries involved in their problem. In tandem with the error now recognised in the supposed defiance of Bell’s inequality by quantum probabilities, my reassessment of their work should motivate a reevaluation of the current consensus outlook regarding the principle of local realism and the proposition of hidden variables.

## 1. Prelude

When I first began to disseminate the results of my analysis that refute the supposed defiance of Bell’s inequality by quantum probabilities [1], I was immediately challenged by suspicious and offended physicists. Considered to be incomprehensible and confused, the focus of my analysis was relegated as being completely out of date with subsequent developments. Not only was Aspect’s experimental setup [2] deemed to be crude and to have been superseded, it was proposed to me that a definitive analysis by Greenberger, Horne, Shimony, and Zeilinger [3] had determined that the premises of the EPR argument entailing “perfect correlation, reality, locality, and completeness” could not even be modeled consistently in quantum problems involving more than two dimensions. GHSZ were said to have expounded a version of Bell’s theorem without involving inequalities. Aware of my evident naiveté regarding matters I had long avoided in my research, and keen to learn what this might be about, I eagerly dove right in!

Upon reading, I was stunned to learn that their claims, already studied and honoured within the physics community for some twenty-five years, are based on an argument that defies a simple rule of deductive logic, and is entrained in a cryptic notational sleight of hand which has misled the leading researchers of the scientific community. Their article will have been read by surely more than five thousand physicists and students during this time. It is with some embarrassment that I shall explain these miscues in this article. I invite you to please continue reading, with an open mind, if only out of curiosity.

Any reader prepared to do so is advised to refresh himself/herself with the GHSZ argument in their own words before continuing. However, to make this presentation self-contained, their argument will be developed here faithfully until the point that it hits the fan, repeating their equations and using their notation precisely until it will become necessary to refine and embellish it. My first Equation (Equation 1) will repeat their equation numbered (7), the first numbered equation of their Section III which is the sole subject matter of my exposition here. Since I do not print their equation numbered (8), my subsequent equation numbers (2–11) repeat theirs, with my equation numbers merely diminishing each of theirs by seven. When I need to repeat the GHSZ equations using a refined notation, their equation numbers 4–6 shall appear as 12,13, and 16 in their new form. Equations numbered 14, 15, and 17 are introduced here as new to the discussion. I mention in passing only that the persuasive exposition of Bell’s original theorem in the Section II of the GHSZ article continues to embed the error which Bell and subsequent advocates of the supposed violation have missed.

My deferential attitude to the reader’s review of the original paper, and the care with which I shall now assess it, are appropriate to the influential role the article has played in motivating the current general understanding of the quantum scale defiance of Bell’s inequality within the physics community. Of course the inequality itself has provoked extensive analysis and a myriad of commentaries. Notable among them are the comprehensive review by Brunner et al. [4] and wide-ranging discussions in the texts of Greenstein and Zajonc [5] and of Jaeger [6]. Within this context, the provocative claims of GHSZ to a contradiction involved in the formulation of theories of local realism are considered to be definitive. They form part of a stimulating literature that stems from the original challenging work of von Neumann [7] and runs through Mermin and Schack [8]. While subscription to various arguments has varied over time, the concluding assessment of GHSZ by Jaeger [6] (pp 44–45), is supported by virtually all researchers whose work is referenced in these reviews:
Remarkably, a decade after Aspect’s tests of the CHSH inequality, it was shown by Greenberger, Horne, Shimony, and Zeilinger (GHSZ) that the premises of the Einstein–Podolsky–Rosen paper become inconsistent when applied to systems possessing three or more subsystems, even for the cases involving such perfect correlations [194]. The GHSZ demonstration shows that the incompatibility of the EPR assumptions with quantum mechanics is stronger than that indicated by the violation of the Bell and CHSH inequalities, in that in the case of a pair of two-level systems there is no internal contradiction at the level of perfect correlations. Indeed, Bell produced an explicit model for the case of a pair of spin-1/2 particles demonstrating the consistency of the EPR conditions with the perfect correlations predicted by quantum mechanics [23]. Furthermore, the contradiction between quantum mechanical predictions and the Bell and CHSH inequalities are expressions violated only by statistical predictions of quantum mechanics, rather than by individual events. In the lead up to the exceptionally clear exposition of GHSZ, Greenberger, Horne, and Zeilinger (GHZ) demonstrated the inconsistency in a new way in systems consisting of three or more correlated spin-1/2 particles [195]. Because this showed that the incompatibility of quantum mechanics with the EPR assumptions arises at the level of perfect correlations rather than statistical predictions and did not require the use of an inequality, these results are often referred to as “Bell’s theorem without inequalities.”


In a word, I am fully aware of the momentous content of the analysis I shall present here, and the extent of the reconsiderations it should require. While my presentation shall be lively, it is not with bravado but with honest concern that I suggest we have been deeply engaged in a serious mistake for some fifty years since the advent of Bell’s first publications on the matter, a mistake he himself had suspected but could not identify. If I am in error it will need to be displayed by reasoning rather than by reverence for vaunted old understandings. We all make mistakes. Let us get into it.

## 2. The Physical Setup of the 4-D GHSZ Experiment

The context of the GHSZ experiment does not involve the behaviour of photons, but rather of four electrons that are propelled toward Stern–Gerlach analysing magnets so to determine the directions of their orbital spins. The physical setup is caricatured by their Figure 2 which is reproduced as our Figure 1.

This schema represents a physical process in which one pair of electrons (referred to as “spin-1/2 particles”) are transmitted in spatially separated beams in the direction $z^{+}$ toward Stern–Gerlach analysers at stations 1 and 2 while another pair are transmitted far away in the opposite direction $z^{-}$ toward similarly separated analysers at stations 3 and 4. The electrons are said to be in a superposition of two states vis-à-vis their spins, implying that they might be observed as either “up” or “down” when detected at any analyser. As transmitted, they are considered to be in both and neither states, superposed. The observation “up” is designated by a recording of +1, while “down” is recorded as −1. Now the Stern–Gerlach magnets can be positioned in various directions relative to that in which the incoming particles arrive. These directional vectors are denoted in the Figure as ${\hat{n}}_{1},{\hat{n}}_{2},{\hat{n}}_{3}$, and ${\hat{n}}_{4}$. In the (x,y) plane perpendicular to any z direction, the Figure denotes by ϕ the angle relative to the *x*-dimension in which any one of them is directed. The four angle sizes $\phi_{1}$, $\phi_{2}$, $\phi_{3}$, and $\phi_{4}$ can be variable, but they are fixed at specific values for any run of the experiment. Of course the (x,y) planes of these detectors could also be twisted toward the direction of their z axes as well. Such can be accounted for in the theory according to a general equation which GHSZ numbered as (8) in which these angles are denoted by $\theta_{i}$. However, they are not depicted in the Figure, nor is this equation displayed here, for throughout this discussion these twist angles are considered to be fixed at $\theta_{i}=\pi /2=90^{\circ }$, each of the four (x,y) planes being perpendicular to the direction of z. This reduces their specific equation numbered (9) to our Equation (2) which we shall examine shortly.

The 4-component electron propagation mechanism, which we shall not discuss here, ensures the spin state of the four-plex of electrons is superposed according to the prescription
(1)$$\;|\Psi \rangle \;\;=\;\;\left(1/\sqrt{2}\right)\;\left[{\;|+\rangle }_{1}{\;|+\rangle }_{2}{\;|-\rangle }_{3}{\;|-\rangle }_{4}{\;\;+\;\;|-\rangle }_{1}{|\;|-\rangle }_{2}{|\;|+\rangle }_{3}{|\;|+\rangle }_{4}\;\right]\;.$$

If you are not au fait with the mathematical apparatus of quantum theory, please do not be put off by this notation, and read on. This is merely a concise notation for representing the structure of the experimental situation we have explained in the previous paragraph. Trust yourself. You are ready to go.

### 2.1. Identifying the Prescriptions of Quantum Theory

When detections are made of the electrons’ spins at stations 1,2,3, and 4, these are designated by the variable names A,B,C, and *D*, respectively. Each of these values might arise as −1 or +1 according to the direction of the corresponding spin observation. Now what would be the values of these observation values in any run of the experiment? The theory of quantum mechanics does not specify exactly what the results would be in any run of such experiments, even though each run in a sequence might be set up carefully in exactly the same way. Quantum theory yields only a probability distribution for the four measurement results from any run. Now just as our investigation of the polarised photon pair detection in the Aspect/Bell experiment found the crucial feature of the observations was the value of the *product* of the detection results, the crucial part of the QM specifications of probabilities for observation values of the four electron spins in a run revolves about their product: ABCD.

The product of the spin measurements in any run depends stochastically on the four angle settings in the design of that run. GHSZ derive the mathematical expectation of this spin-product in Appendix F of their article for the general case in which the angles $\theta_{i}$ are variable. In the article itself they simplify this expectation to the special case in which each of the (x,y) planes is perpendicular to its relevant *z* axis. Using cryptic notation, the spin-product expectation is reported for this setup in an equation denoted as
(2)$$\;E^{\Psi }\left({\hat{n}}_{1},{\hat{n}}_{2},{\hat{n}}_{3},{\hat{n}}_{4}\right)\;=\;-cos\left(\phi_{1}+\phi_{2}-\phi_{3}-\phi_{4}\right).$$

They describe this equation as, “The expectation of the product of the outcomes when the orientations are as indicated.” To be sure, this notation has a sensible motivation. For it pertains to an expectation of a function (the product) of spin observations on a system of particles designed to reside in the superposed state designated by |Ψ〉 in which the Stern–Gerlach analysers are positioned at angles corresponding to the directional vectors ${\hat{n}}_{1},{\hat{n}}_{2},{\hat{n}}_{3}$, and ${\hat{n}}_{4}$. These directional vectors determine the angles $\phi_{1},\phi_{2},\phi_{3}$, and $\phi_{4}$ whose linear combination is the argument of the cosine function in (2) that specifies the expectation value. Well, I do not think I am being picayune, and I hope I am not, in highlighting that this expectation $E^{\Psi }\left({\hat{n}}_{1},{\hat{n}}_{2},{\hat{n}}_{3},{\hat{n}}_{4}\right)$ is *not* the expectation of a vector of four directional vectors (which it ostensibly is), but rather it is the expectation of the product of four spin observation values in an experiment at which the analysing magnets are aligned with these directions. It would be depicted more clearly as identifying $E^{\Psi }\left[ABCD\left(\phi_{1},\phi_{2},\phi_{3},\phi_{4}\right)\right]$. While understandable, the notation they use serves to obscure the sensible analysis of the situation, as we shall now discover.

Of particular interest to the GHSZ argument are the cases of perfect correlation, which they designate by the conditioned equations
(3a)$$\begin{array}{c} \mathrm{If}\;\phi_{1}+\phi_{2}-\phi_{3}-\phi_{4}\;=\;0\\ \mathrm{then}\;E^{\Psi }\left({\hat{n}}_{1},{\hat{n}}_{2},{\hat{n}}_{3},{\hat{n}}_{4}\right)\;=\;-1\;, \end{array}$$
(3b)$$\begin{array}{c} \mathrm{If}\;\phi_{1}+\phi_{2}-\phi_{3}-\phi_{4}\;=\;\pi \\ \mathrm{then}\;E^{\Psi }\left({\hat{n}}_{1},{\hat{n}}_{2},{\hat{n}}_{3},{\hat{n}}_{4}\right)\;=\;+1\;. \end{array}$$

The conditions are printed exactly like that, within four lines of which the second and fourth lines conclude with the equation number designations (3a) and (3b). This unfortunate feature of the double column print style of their publication will be shown to be relevant to their mistaken argument. A second unfortunate feature of their considerations is that they did not designate explicitly in notation the statement that they described in words: that the single observed product ABCD of the spin recordings depends (stochastically) on the *four* orientations of the detection magnets, $\left(\phi_{1},\phi_{2},\phi_{3},\phi_{4}\right)$. Mentioning for now only that these two misfortunes will return to haunt us, we shall continue with the GHSZ argument.

Interest in the conditions of (3a) and (3b) arises from the fact that they support the supposition of Einstein, Podolsky, and Rosen [9] of perfect correlation, a condition specified by the derivations of quantum theory. The other three suppositions of EPR are noted to concern matters extraneous to quantum theory, though they are recognised as plausible and in agreement with principles of classical physics.

GHSZ were pleased that this experimental design might instantiate the requirements of Bell’s inequality in a four-dimensional problem. However, they proceeded no further in pursuing details of this matter, because they first pursued an argument that the structure developed to this point embeds a contradiction among the four premises of EPR, these being (i) perfect correlation, (ii) reality, (iii) locality, and (iv) completeness. Let us begin to follow their line of argument.

### 2.2. Pursuing the Contradiction Discovered by GHSZ

They begin by restating the conditional statements (3a) and (3b) but now using functional observation notations $A{(}^{.}),\;B{(}^{.}),\;C{(}^{.})$, and $D{(}^{.})$ in stating the conclusions of the “if” clauses. They write
(4a)$$\begin{array}{c} \mathrm{If}\;\phi_{1}+\phi_{2}-\phi_{3}-\phi_{4}\;=\;0\\ \mathrm{then}\;A_{\lambda }\left(\phi_{1}\right)B_{\lambda }\left(\phi_{2}\right)C_{\lambda }\left(\phi_{3}\right)D_{\lambda }\left(\phi_{4}\right)\;=\;-1\;, \end{array}$$
(4b)$$\begin{array}{c} \mathrm{If}\;\phi_{1}+\phi_{2}-\phi_{3}-\phi_{4}\;=\;\pi \\ \mathrm{then}\;A_{\lambda }\left(\phi_{1}\right)B_{\lambda }\left(\phi_{2}\right)C_{\lambda }\left(\phi_{3}\right)D_{\lambda }\left(\phi_{4}\right)\;=\;+1\;. \end{array}$$

Before continuing with their development, I must make two remarks, either of which might be ignored, but both of which are of some meritorious consequence.

The first concerns an unspoken argument. Notice that the lines numbered (3a) and (3b) make statements about the values of *expectations of the products* of the spin measurements at the four observation sites, whereas lines (4a) and (4b) concern the *products of observations themselves*. Upon consideration, this appears to be of little consequence. Why? The numerical values of each multiplicand can be only either −1 or +1 by their very operational definition, so the product of any four such observations may also equal only either −1 or +1. Moreover, Equation (Equation 3), for example, quite rightly designates that under the condition specified in the unnumbered line just above it, the expectation must equal −1. This implies that the probability weight attributed to the possibility that the product equals +1 must be zero! In this condition, the product of the spins itself must equal −1. Thus, quantum theory prescribes that in this case the *product* of the four spin observations itself must equal −1, just as stated in Equation (Equation 4). The very same remark would pertain to the replacement of Equations (Equation 3) by (4b), according to which the product of the four observations must equal +1 under the condition to which it pertains.

So changing the statement of Equation (3a,b) regarding expectations of a function (the product of four measurements) to Equation (4a,b) regarding the function values themselves appears innocuous here, because the distributions of the function value are obviously degenerate in both cases, at −1 and +1 respectively. However, we should be aware that the joint distributions of the component multiplicands of this product have *not* been determined to be degenerate. There are several arrays of the multiplicands that could yield a product of −1 to instantiate Equation (Equation 4), and several arrays that could yield a product of +1 so to instantiate Equation (Equation 4). It is the expectation of their product which equals −1, and thus their product which has a degenerate distribution. The distribution of the four component observations in a run remains ample. Enough said for now, but we shall return to this recognition later.

The second remark is less consequential, but it is substantive relevant to the larger picture of what this analysis of the GHSZ problem is all about. At this point in their development, the spin observation variables A,B,C, and *D* suddenly begin to appear with the subscript λ, as in $A_{\lambda }\left(\phi \right)$. As it turns out, *every* subsequent appearance of such observation values in their article is adorned with this subscript. Thus, the subscript contributes nothing to the force of the relevant arguments, and I will no longer use it. To be sure, its presence would serve as a reminder that what we are doing here is formalising experimental matters motivated by considerations of the EPR proposal of “supplementary variables” as the source of the incompleteness of quantum theory. The vector λ is meant to designate the unknown values of such variables. Now this *is* a very important matter bearing consideration, and I am going to investigate this matter explicitly and on its own in a separate article. However for now, let us just consider ourselves to have been reminded of this important motivation for the considerations we are pursuing, and just drop the λ subscript, except in places where we will quote GHSZ verbatim, for there will come a point in these considerations at which we shall wish to use the subscript position on the spin measurements to make explicit an important varying feature of the problem that has been ignored hitherto.

Let us continue then from Equation (4a,b). I shall be following very closely the work and even the wording of GHSZ. When exact quotations are used they will be marked.

Next are considered some implications of (4a). Four specific instances of this specification are proposed, with reference to any arbitrary angle size ϕ, as
(5a)A(0)B(0)C(0)D(0)=−1
(5b)A(ϕ)B(0)C(ϕ)D(0)=−1
(5c)A(ϕ)B(0)C(0)D(ϕ)=−1
(5d)A(2ϕ)B(0)C(ϕ)D(ϕ)=−1,
because the four angle arguments in each of these product equations meet the condition that $(\phi_{1}+\phi_{2}-\phi_{3}-\phi_{4})=0$, which is the condition under which (4a) holds.

As a consequence of Equalitie (5a,b), they then obtain
(6a)A(ϕ)C(ϕ)=A(0)C(0),
through cancellation of B(0)D(0) which appears identically on the left-hand sides of these two equations. Correspondingly, equalities (5a) and (5c) are seen to imply
(6b)A(ϕ)D(ϕ)=A(0)D(0)
according to similar cancellations of B(0)C(0).

The quotients of these two results then yield for them (7a)C(ϕ)/D(ϕ)=C(0)/D(0),
which can be rewritten as
(7b)C(ϕ)D(ϕ)=C(0)D(0),
because both D(ϕ) and D(0) can each equal only either +1 or −1. In either case, both of them are equal to their inverses.

By substituting (7b) into (5d), GHSZ then obtain the result that (8)A(2ϕ)B(0)C(0)D(0)=−1.


This, in combination with (12a) which says A(0)B(0)C(0)D(0)=−1, yields (9)A(2ϕ)=A(0)=constforanyangleϕ

In particular, this would imply that a measurement of A(π) must equal A(0).

While this result is apparently not contradictory, to a physicist well versed in quantum theory it is surely quite troublesome. GHSZ call it a “surprising preliminary result.” Let them explain why it appears troublesome in their own words. “For if $A_{\lambda }\left(\phi \right)$ is intended, as EPR’s program suggests, to represent an intrinsic spin quantity, then $A_{\lambda }\left(0\right)$ and $A_{\lambda }\left(\pi \right)$ would be expected to have opposite signs.” (Be aware that an electron spin measured by analysers positioned in two opposing directions, differing in orientation by $\pi =180^{\circ }$, should display opposite directions rather than the same direction, just as two attracting magnets will repel one another if one of them is rotated by $180^{\circ }$.) However, this trouble is subjected to no further examination, on account of a really stunning argument.

### 2.3. Witness the Sleight of Hand!

GHSZ continue (using our numberings in their quotation): “The trouble becomes manifest, and an actual contradiction emerges, when we use (4b)—which until now has not been brought into play—to obtain
(10)$$A_{\lambda }\left(\theta +\pi \right)B_{\lambda }\left(0\right)C_{\lambda }\left(\theta \right)D_{\lambda }\left(0\right)\;=\;1$$
which in combination with Equation (Equation 5) yields (11)$$A_{\lambda }\left(\theta +\pi \right)\;=\;-A_{\lambda }\left(\theta \right)\;\;.$$


“This result *confirms* the sign change that we anticipated on physical grounds in EPR’s program, but it also *contradicts* the earlier result of Equation (9) (let ϕ=π/2,θ=0). We have thus brought to the surface an inconsistency hidden in premises (i)–(iv).” Their parenthetical suggestion means to let θ=0 in (11) and to let ϕ=π/2 in (9).

Wow! What could be a more stunning and insightful demolition of the EPR premises?

Answer: some calm logical thinking, and the truth! Well, what could be wrong here?

### 2.4. A Little More Denken

To begin with a startling observation, neither of the equations numbered (4a) and (4b) stands on its own. Both of them are conclusions of a conditioning clause, an “if clause”; and these two clauses are quite evidently contradictory. Yet GHSZ use these two conclusions in concert. Equation (Equation 4) results from quantum theory applied to a situation in which the four magnet angles are designed to meet the condition, “If $\phi_{1}+\phi_{2}-\phi_{3}-\phi_{4}\;=\;0$.” Equation (Equation 4) provides the conclusion to the condition, “If $\phi_{1}+\phi_{2}-\phi_{3}-\phi_{4}\;=\;\pi$.” If either of these conditions holds for an experiment under consideration then the other cannot. The two conditions can not be instantiated at the same time, and they may not be honoured at the same time—nor may the distinct conclusions they motivate. It is surely not permitted to combine their Equation (10) with Equation (5b) to yield (11). Professors Greenberger, Horne, Shimony, and Zeilinger discover the contradiction in the EPS suppositions that they do *because they have introduced the contradiction into their analysis themselves!* Full stop.

Compounding the misunderstanding this analysis has supported, empirical results from an actual experiment in a companion context of three photon polarisations was presented by Pan et al. [10] claiming to prove the inadequacy of a locally realistic model to account for them. This was challenged by Aschwenden et al. [11] who proposed just such a model that improved the experimental explanation provided by a purely quantum theoretic model. In fact, problems raised by the empirical programme engaged by Pan et al. run even deeper than this. To avoid a distraction here I comment only briefly (explicitly enough only for readers acquainted with details of this highly regarded publication) in parentheses. (The empirical programme of Pan et al. [10] who include Zeilinger relies on two erroneous presumptions. In the first place, the joint activation of three different polarisation designs $Y_{1}Y_{2}X_{3},Y_{1}X_{2}Y_{3},$ and $X_{1}Y_{2}Y_{3}$ cannot be performed on the same triplet of photons, so in deference to the general uncertainty principle, quantum theory explicitly avoids any claims regarding their simultaneous result. It is a “thought experiment” to which the implications of locality would pertain, asserting that the value of $Y_{2}$, for example, in a simultaneous instantiation of the $Y_{1}Y_{2}X_{3}$ design must be identical to its value in an $X_{1}Y_{2}Y_{3}$ design activation on the same triple of photons. This is a claim widely recognised as lying outside the domain of quantum theoretical assertions. In the second place, in their empirical evaluation of results on three *different* triples of photons, the conditions of quantum superposition assure only that the *product* values of $Y_{1}Y_{2}X_{3}$ and $X_{1}Y_{2}Y_{3}$ are both identically equal to −1, not that any individual multiplicands of the product triples are equal. In particular, the value of $Y_{2}$ in the first experimental triple need not equal the value of $Y_{2}$ in the second triple. The implication proposed in their analysis that the value of $X_{1}X_{2}X_{3}$ in any triple run on three distinct triples of photons that meet the GHZ condition in their experiment does not hold. Their allusion to experimental error as accounting for their mixed results does not wash. I would be happy to be more explicit in discussion or in another article, but I will leave these comments in this terse form for now.)

Returning to our assessment of the GHSZ article itself, recognising the joint supposition of the contradictory premisses to (4a) and (4b) alone should have startled any serious reader. It should surely not allow the acclaim that has been afforded to the spurious result of GHSZ. However, the confusions in their argument run even deeper still. There is much more that can be learned by a continued pursuit of the surprising and troublesome preliminary result (9) from which their supposed discovery of a contradiction then deterred them. We shall learn now that this very result itself derives from another serious error. Less evident to a new reader, perhaps, it is nonetheless shocking, having arisen from insufficient thinking combined with the use of casual notation. Let us look into it.

### 2.5. Completing the Notation, and Continuing Pursuit of Trouble

I had initially been taken in by the GHSZ argument, and I shared their puzzlement. However, whereas they were concerned with the identical signs of a spin observed from opposite directions, I wondered how the angle $\phi_{1}$ at station *A* might equal both 2ϕ and equal 0 so as to instantiate Equation (9). The quantities $A(2\phi_{1})$ and A(0) explicitly denote spin observations at station 1 in two different experiments in which the orientation of the detection magnet differ. In either of them the outcome of the experiment is random, specified by quantum probabilities, but allowing each of them to equal −1 or +1. There is no requirement of any sort that these outcomes need be identical in the two different experimental runs. What could their equation A(π)=A(0) mean? It turns out that there is a clear way out of either conundrum, as we shall see. We shall reconsider their development of (9) and find how to clarify the situation. Sad to say it though I am, we shall need to start at the beginning.

The GHSZ argument was proposed some fifty years after the recognition of particle entanglement had arisen among quantum theorists. This had made all of us aware that consideration of any aspect of particle behaviour observed at station 1 with a measurement labelled *A* will typically depend on both the settings of the angles at the other three stations, and depend on the behaviour of the particles observed there as well. I mention this because at the very start of their argument, GHSZ casually denote the particular instantiations of their expectation Equation (3a,b) by writing (4a) and (4b) as $$\begin{array}{c} \mathrm{If}\;\phi_{1}+\phi_{2}-\phi_{3}-\phi_{4}\;=\;0\\ \mathrm{then}\;A_{\lambda }\left(\phi_{1}\right)B_{\lambda }\left(\phi_{2}\right)C_{\lambda }\left(\phi_{3}\right)D_{\lambda }\left(\phi_{4}\right)\;=\;-1\;,\\ \mathrm{If}\;\phi_{1}+\phi_{2}-\phi_{3}-\phi_{4}\;=\;\pi \\ \mathrm{then}\;A_{\lambda }\left(\phi_{1}\right)B_{\lambda }\left(\phi_{2}\right)C_{\lambda }\left(\phi_{3}\right)D_{\lambda }\left(\phi_{4}\right)\;=\;+1\;. \end{array}$$

Equation (3a,b) had quite rightly designated the expectations in their concluding clauses (the left-hand sides of the equations in their “then” statements) as functions of four directional variables, as in $E^{\Psi }\left({\hat{n}}_{1},{\hat{n}}_{2},{\hat{n}}_{3},{\hat{n}}_{4}\right)$. Although I had remarked about the cryptic form of this notation, their verbal description of it did have the feature of recognising that this is an expectation of a general function of four variables, the directional vectors of the four magnet settings. However, in their instantiation Equations (4a,b) GHSZ blithely represent this product function $ABCD(\phi_{1},\phi_{2},\phi_{3},\phi_{4})$ as a separable function of four independent variables standing alone in singular directional settings: $A\left(\phi_{1}\right)B\left(\phi_{2}\right)C\left(\phi_{3}\right)D\left(\phi_{4}\right)$. For the moment I say only, “Beware!” But this misconstrual of the situation becomes even more abusive, and we shall need to look into it deeply to uncover the havoc it has promoted.

A complete designation of the arguments of these spin measurement functions A,B,C, and *D* needs be made to denote fully the experimental context in which the observations are made. The measurement which GHSZ denote as $A_{\lambda }\left(\phi_{1}\right)$ requires embellishment to $A_{\lambda }\left(\phi_{1},\phi_{2},\phi_{3},\phi_{4}\right)$ if it is to denote an observation of the spin *A* on the first of four entangled particles. The magnet angle at station 1 may well have been set at $\phi_{1}$, but what were the directional angles for the spin detectors of the other three particles with which the particle entering station 1 is entangled? This surely makes a difference, as the distinction between Equation (3a,b) makes clear. On the one hand, one might might insist on the full experimental design notation $A(\phi_{1},\phi_{2},\phi_{3},\phi_{4})$ in the identification of the spin measurement *A* in any experimental run. However this would amount to an ungainly designation of a spin-product of four such measurements. I am going to suggest and to follow henceforth a notation that will use the subscript position under *A* to designate the full angular context in which a measurement of *A* at its station angle $\phi_{1}$ is made. So we will write $A_{\left(\phi_{1},\phi_{2},\phi_{3},\phi_{4}\right)}\left(\phi_{1}\right)$ to designate the full angular context in which a measurement of *A* at its station angle $\phi_{1}$ might be made. Remember that we are forgoing the unvarying GHSZ universal subscript of λ on spin values A,B,C, and *D*, except at times when I quote them exactly.

Using this notational convention we need to rewrite the GHSZ functional form Equation (4a,b) of the general expectation result (3a,b), in the form
(12a)$$\mathrm{If}\;\phi_{1}+\phi_{2}-\phi_{3}-\phi_{4}\;=\;0\;\mathrm{then}\;A_{\left(\phi_{1},\phi_{2},\phi_{3},\phi_{4}\right)}\left(\phi_{1}\right)B_{\left(\phi_{1},\phi_{2},\phi_{3},\phi_{4}\right)}\left(\phi_{2}\right)C_{\left(\phi_{1},\phi_{2},\phi_{3},\phi_{4}\right)}\left(\phi_{3}\right)D_{\left(\phi_{1},\phi_{2},\phi_{3},\phi_{4}\right)}\left(\phi_{4}\right)\;=\;-1\;,\;\;\mathrm{and}$$
(12b)$$\mathrm{If}\;\phi_{1}+\phi_{2}-\phi_{3}-\phi_{4}\;=\;\pi \;\mathrm{then}\;A_{\left(\phi_{1},\phi_{2},\phi_{3},\phi_{4}\right)}\left(\phi_{1}\right)B_{\left(\phi_{1},\phi_{2},\phi_{3},\phi_{4}\right)}\left(\phi_{2}\right)C_{\left(\phi_{1},\phi_{2},\phi_{3},\phi_{4}\right)}\left(\phi_{3}\right)D_{\left(\phi_{1},\phi_{2},\phi_{3},\phi_{4}\right)}\left(\phi_{4}\right)\;=\;+1\;.$$

Apologies for this complex notation, but we shall require it to air an egregious error in the GHSZ argument, to which we now turn.

Appearing now *somewhat* less ungainly, the specific instantiation equations they enumerate as Equation (5a–d) would be designated by (13a)$$A_{\left(0,0,0,0\right)}\left(0\right)B_{\left(0,0,0,0\right)}\left(0\right)C_{\left(0,0,0,0\right)}\left(0\right)D_{\left(0,0,0,0\right)}\left(0\right)\;\;=\;\;-1$$
(13b)$$A_{\left(\phi ,0,\phi ,0\right)}\left(\phi \right)B_{\left(\phi ,0,\phi ,0\right)}\left(0\right)C_{\left(\phi ,0,\phi ,0\right)}\left(\phi \right)D_{\left(\phi ,0,\phi ,0\right)}\left(0\right)\;\;=\;\;-1$$
(13c)$$A_{\left(\phi ,0,0,\phi \right)}\left(\phi \right)B_{\left(\phi ,0,0,\phi \right)}\left(0\right)C_{\left(\phi ,0,0,\phi \right)}\left(0\right)D_{\left(\phi ,0,0,\phi \right)}\left(\phi \right)\;\;=\;\;-1$$
(13d)$$A_{\left(2\phi ,0,\phi ,\phi \right)}\left(2\phi \right)B_{\left(2\phi ,0,\phi ,\phi \right)}\left(0\right)C_{\left(2\phi ,0,\phi ,\phi \right)}\left(\phi \right)D_{\left(2\phi ,0,\phi ,\phi \right)}\left(\phi \right)\;\;=\;\;-1\;,$$
for the components of the subscripted angle configuration vectors underlying each of these product designations meet the condition that $\phi_{1}+\phi_{2}-\phi_{3}-\phi_{4}=0$, the condition under which (12a) holds.

As an appeasement to your understandable hopes for a simpler notation, I might mention only that there will be some contexts in which it will be sufficient to subscript a spin observation by the value of $\kappa =\phi_{1}+\phi_{2}-\phi_{3}-\phi_{4}$, so to write something like $A_{\kappa }\left(\phi_{1}\right)=A_{0}\left(0\right)$ in place of $A_{\left(0,0,0,0\right)}\left(0\right)$. However in the four lines of (13), the experimental contexts are such that every spin observation would then be subscripted with 0. This would mean, for example, that $D_{\left(0,0,0,0\right)}\left(0\right)$ in (13a) and $D_{\left(\phi ,0,\phi ,0\right)}\left(0\right)$ in (13b), would both be designated by $D_{0}\left(0\right)$, whereas in fact they quite rightly designate very different things, these being the observation values of *D* in two quite different experiments and experimental settings. There is no assurance at all that they will instantiate at the same numerical value, as we shall now see. GHSZ casually ignore this whole contextual matter about which their knowledge of quantum entanglement should have alerted them. They designate them both by $D_{\lambda }\left(0\right)$, and presume that they are always equal in any run of their experiment, feeling free to cancel them against one another when they do their algebra. Then they are briefly surprised by their Equation (9) which results. However, as they carry on with the self-contradictory analysis we have discussed, they then ignore the entire issue, surprising preliminary result and all. To the contrary, we shall now trudge into these matters in great gory detail, but we shall first provide a clarifying assessment of the entire setup.

### 2.6. Clarification via Examination of the Realm Matrices

For clarification of a central aspect of the situation relevant to all that follows, my Equation (14) displays the ensemble of observation possibilities (which I call a “realm matrix”) for an observable spin vector arising in the conduct of a specific Stern–Gerlach experiment on four entangled particles. The observations are made on four particles in a quantum state |Ψ〉 corresponding to *any* specific experimental design for which $\phi_{1}+\phi_{2}-\phi_{3}-\phi_{4}\;=\;0$. To repeat, this condition is a supposition (“if clause”) of GHSZ Equation (Equation 4) which does not stand on its own without this clause. Notice that the companion Equation (Equation 4) relies on an alternative condition that $\phi_{1}+\phi_{2}-\phi_{3}-\phi_{4}\;=\;\pi$. Contradictory to each other, both of these conditions cannot be satisfied in any specific experimental run, nor even in the imagined runs of a gedankenexperiment on a specific single quartet of particles. The angle combination may equal either 0 or π in any experimental setup, but it cannot equal both at the same time; nor can their two concluding equations hold at the same time if we are to insist on discussion that honours the prescriptions of deductive logic. It is a sad commentary on our times that we need to make explicit this proviso.

In displaying the realm matrix below, I denote the specific experimental quantity observation vector in its shorthand form using the subscript κ just mentioned: ${\left[A_{0}\left(0\right),B_{0}\left(0\right),C_{0}\left(0\right),D_{0}\left(0\right)\right]}^{T}$. The subscripts 0=κ on each of the spin observation values denotes the contextual value of the angle combination pertaining to the experiment, and the arguments of the vector components designate that each of the four angles $\phi_{i}$ is equal to 0 radians. However, it would constitute no loss of generality to recognise this realm matrix of possibilities as pertinent to a spin observation vector resulting from any design that meets this four-angle condition, κ=0.

Since this restriction that $\phi_{1}+\phi_{2}-\phi_{3}-\phi_{4}\;=\;0$ implies via Equation (Equation 2) that the expected spin-product $A_{\lambda }\left(0\right)B_{\lambda }\left(0\right)C_{\lambda }\left(0\right)D_{\lambda }\left(0\right)$ equals −1, it would be impossible to achieve any four experimental spin results that allow the vector of these multiplicands to imply a positive product. The quantum probability weight on such observation vectors must equal 0, and such observation vectors would be impossible. (If such an observation *were* seemingly made, the experimental quantum theorist would typically reject it as valid, and check the settings of the direction angles of the four Stern–Gerlach magnets in the experimental run that generated it.) Thus, the realm matrix of possibilities for the column vector of these four-particle measurements is (14)$$R\left(\begin{array}{c} A_{0}\left(0\right)\\ B_{0}\left(0\right)\\ C_{0}\left(0\right)\\ D_{0}\left(0\right) \end{array}\right)\;\;=\;\;\left(\;\begin{array}{cccccccc} 1 & 1 & 1 & -1 & -1 & -1 & -1 & 1\\ 1 & 1 & -1 & 1 & -1 & -1 & 1 & -1\\ 1 & -1 & 1 & 1 & -1 & 1 & -1 & -1\\ -1 & 1 & 1 & 1 & 1 & -1 & -1 & -1 \end{array}\right)\;\;\equiv \;\;R_{-1.}$$

In the entangled state of the four-particle system specified by |Ψ〉 in Equation (7), the columns of this matrix exhaust the vector values of measurements that can arise from such an experiment. The product of any of these eight ensemble column components equals −1.

Notice the concluding *definition symbol*, (≡), introducing the denotation $R_{-1}\;$ at the right end of Equation (i). This is to distinguish it from a companion matrix to be denoted by $R_{+1}$ which specifies the realm matrix corresponding to the possible outcomes of a *different experiment* in which the magnet angles satisfy instead the condition providing for (11b), that $\phi_{1}+\phi_{2}-\phi_{3}-\phi_{4}\;=\;\pi \;$. Again, without loss of generality, an exemplar experiment would generate an observable result ${\left(A_{\pi }\left(\pi \right),B_{\pi }\left(0\right),C_{\pi }\left(0\right),D_{\pi }\left(0\right)\right)}^{T}\;$, with a realm matrix of columns whose products all equal +1:
(15)$$R\left(\begin{array}{c} A_{\pi }\left(\pi \right)\\ B_{\pi }\left(0\right)\\ C_{\pi }\left(0\right)\\ D_{\pi }\left(0\right) \end{array}\right)\;\;=\;\;\left(\begin{array}{cccccccc} 1 & -1 & -1 & -1 & 1 & 1 & 1 & -1\\ 1 & -1 & 1 & 1 & -1 & -1 & 1 & -1\\ 1 & 1 & -1 & 1 & -1 & 1 & -1 & -1\\ 1 & 1 & 1 & -1 & 1 & -1 & -1 & -1 \end{array}\right)\;\;\equiv \;\;R_{+1}.$$

The restrictions of the measurement vector possibilities embedded in the realm matrices $R_{-1}$ and $R_{+1}$ derive from Equation (Equation 2) of GHSZ, which specifies that $E^{\Psi }\left({\hat{n}}_{1},{\hat{n}}_{2},{\hat{n}}_{3},{\hat{n}}_{4}\right)\;=\;-cos\left(\phi_{1}+\phi_{2}-\phi_{3}-\phi_{4}\right)$. At the two extreme angle restrictions we have entertained with $\phi_{1}+\phi_{2}-\phi_{3}-\phi_{4}$ equal 0 or π, this negative cosine value equals −1 and +1 respectively. This is what restricts the measurement realms to be $R_{-1}$ and $R_{+1}$ in these extreme cases. If the combination of experimental angles in this equation were to equal some other value, say, κ∈(0,π), then the realm matrix of possibilities for the measurements of the four electron spins would be the concatenation of these two realms, $\left[R_{-1}\;R_{+1}\right]$. With the recognition of this situation clearly in mind, we are ready to face the wall.

### 2.7. Hitting a Wall at Equation (9)? Can We Get There? Where Do We Arrive?

GHSZ regard their conclusion (9), that $A_{\lambda }\left(2\phi \right)=A_{\lambda }\left(0\right)=const$ for any angle ϕ, as surprising, for reasons they have well explained. Thinking that nonetheless this equation is not mathematically contradictory in itself, they were sidetracked into the mistaken analysis that presumes jointly the contradictory suppositions of their full two-lined expressions of (4a) and (4b). Having “found” a contradiction in their results, they quit.

Having recognised their error, we shall continue our investigation of their conundrum. We have already recognised that their Equation (9) derives from analysis that uses incomplete notation for describing the situation. Using our completed notation, we have embellished their statement of Equations (4) and (5) to the formulation in (12a,b) and (13a–d). Let us now try to follow their line of argument using our complete notation and see whether we will still hit the wall at their (9).

#### 2.7.1. Heading Toward the Wall

As a consequence of Equalitie (5a,b), GHSZ obtained A(0)B(0)C(0)D(0)=A(ϕ)B(0)C(ϕ)D(0),
which allowed them to conclude their Equation (Equation 6) asA(ϕ)C(ϕ)=A(0)C(0),
through cancellation of B(0)D(0) which appears identically on both sides of that equation. However, we find that by writing the Equation (5a,b) in full notation that includes the contextual subscripts, the Equalitie (13a,b) and obtain for us the equation
(16)$$A_{\left(0,0,0,0\right)}\left(0\right)B_{\left(0,0,0,0\right)}\left(0\right)C_{\left(0,0,0,0\right)}\left(0\right)D_{\left(0,0,0,0\right)}\left(0\right)\;\;\;\;=\;\;A_{\left(\phi ,0,\phi ,0\right)}\left(\phi \right)B_{\left(\phi ,0,\phi ,0\right)}\left(0\right)C_{\left(\phi ,0,\phi ,0\right)}\left(\phi \right)D_{\left(\phi ,0,\phi ,0\right)}\left(0\right)$$


In full notation, the multiplicands that appeared identically as B(0)D(0) on the left-hand sides of (5a) and (5b), now appear as two quite different things in (13a) and (13b)—as they should! For they represent the products of the *B* and *D* spins observed in two completely different experiments: $B_{\left(0,0,0,0\right)}\left(0\right)D_{\left(0,0,0,0\right)}\left(0\right)$ and $B_{\left(\phi ,0,\phi ,0\right)}\left(0\right)D_{\left(\phi ,0,\phi ,0\right)}\left(0\right)$.

It is true that the products of all four subscripted experimental settings on the two sides of this equation must equal −1. However, there is no restriction at all on the product of any *two* of them. Examining the realm matrix $R_{-1}$ it is clear that the product $B_{0}\left(0\right)D_{0}\left(0\right)$ may equal either −1 or +1 in any experimental run for which the directional angle setting specifies κ=−1. Evaluate the component-wise products of rows 2 and 4 of Equation (14) to see that the elements of the resulting product row alternate between −1 and +1. In any two such runs there is no assurance that the value will be the same. We are not permitted to cancel the terms $B_{\left(0,0,0,0\right)}\left(0\right)D_{\left(0,0,0,0\right)}\left(0\right)$ and $B_{\left(\phi ,0,\phi ,0\right)}\left(0\right)D_{\left(\phi ,0,\phi ,0\right)}\left(0\right)$ from the two sides of our Equation (16). They are relevant to the observations of two distinct experiments, and either, both, or neither of them might be equal to −1. The spin-product combination κ of all four subscripted angles specifies only whether the expectation of the product of *all four* spin values must equal −1 or not.

The argument of GHSZ that obtains their Equation (Equation 6) is specious. It does involve a more sophisticated error than their simple error of contradictory logic which has already been castigated. But it is a seriously consequential error nonetheless, an error of insufficient thought applied to casual notation. Continuing with this investigation, we find GHSZ repeat this error in their “derivation” of (6b), leading to a real fiasco in their development of Equation (7a,b), their interim result of (8), and their “surprising preliminary result” of (9). Their “troublesome” conclusion to it all, that A(π) must equal A(0), can now be dismissed as gibberish. We never reach the wall at all!

#### 2.7.2. Where Have We Arrived?

Well, what is to be made of the considerations of GHSZ? We had followed them through their Equations (2) and (3a,b), albeit our allusion to their use of cryptic notation. We had proposed that the conclusion of their analysis through (3) would be expressed more clearly as $E^{\Psi }\left[ABCD\left(\phi_{1},\phi_{2},\phi_{3},\phi_{4}\right)\right]\;=\;-cos\left(\phi_{1}+\phi_{2}-\phi_{3}-\phi_{4}\right)$. The entanglement among the four multiplicand spins is evident. Moreover, it is readily apparent that this conclusion displays a symmetry of the quantum spin-product expectation with respect to rotations of the entire 4-ply Stern–Gerlach mechanism in the (x,y) dimensions at the four observation stations, and more! For as long as $t_{1}+t_{2}=t_{3}+t_{4}$, it is evident that
(17)$$E^{\Psi }\left[ABCD\left(\phi_{1}+t_{1},\phi_{2}+t_{2},\phi_{3}+t_{3},\phi_{4}+t_{4}\right)\right]\;=\;E^{\Psi }\left[ABCD\left(\phi_{1},\phi_{2},\phi_{3},\phi_{4}\right)\right],$$
because the value of κ associated with both of these 4-angle designs is identical. Let us think.

#### 2.7.3. Recognising Rotational as Well as Permutation Symmetry

Algebraically, two types of symmetry are evident in the QM-motivated stipulations of Equation (2), which shows that the expected 4-spin-product is a function only of the angle combination $\phi_{1}+\phi_{2}-\phi_{3}-\phi_{4}\;$. Rotational symmetry of the experimental conditions would be exhibited in the transformation of the vector of angles $\left(\phi_{1},\phi_{2},\phi_{3},\phi_{4}\right)$ by the addition of any vector of constants (t,t,t,t), which would surely preserve the angle combination κ. In fact, preservation would continue under the addition of any angle vector $t_{4}$ for which $t_{1}+t_{2}=t_{3}+t_{4}$. This general condition allows permutations of either or both of $\phi_{1}$ with $\phi_{2}$ and/or $\phi_{3}$ with $\phi_{4}$ in the specification of κ. It would also allow the permutation of any pair $\left(\phi_{1},\phi_{2}\right)$ with $\left(\phi_{3},\phi_{4}\right)$, because the cosine of any angle is identical to the cosine of the negative of that angle. Geometrically, rotational symmetry would allow rotation of the (x,y) planes containing the directional vectors $\left({\hat{n}}_{1},{\hat{n}}_{2},{\hat{n}}_{3},{\hat{n}}_{4}\right)$ in Figure 1 around the (x,y) axes orthogonal to the −z↔z axes, all to the same degree. Permutation symmetry would allow the exchange of the (x,y) axes of directional designations between stations 1 and 2 or between 3 and 4, or even an exchange of these axis systems of stations (1,2) with those of (3,4). Moreover, the invariance with respect to $t_{1}+t_{2}=t_{3}+t_{4}$ is even richer than all these symmetries, because it allows the twisting of the (x,y) axes at station 1 and station 2 in any ways one wishes, just so long as one concomitantly twists the axes at station 3 and station 4 correspondingly, ensuring that the *sum* of the two latter twisting degrees equals the *sum* of the twisting degrees of the former two.

A final comment on this situation is in order before we consider a Bell inequality formulation in the context of the 4-dimensional GHSZ experiment proposal. Notice that the QM-motivated invariance Equation (17) specifies the invariance of the expectation of the *product* of the four spin observations with respect to all these forms of twisting any initial reference 4-angle configuration. However, Equation (17) does *not* specify invariance of the component multiplicands themselves that generate the product. Each time we suggest a twisting of the four Stern–Gerlach magnet orientations in the (x,y) planes at the four stations—however we propose to do it—we are considering a new distinct run of the 4-ply spin experiment of GHSZ. Even in the special cases in which the initial reference configuration specifies κ to be equal to 0 or π, the invariance pertains to the product of the multiplicand spin observations, not to the multiplicand spin values themselves.

This is what quantum theory tells us about the 4-dimensional experiments that GHSZ propose. We have not yet concerned ourselves with the claims regarding local realism and hidden variables, the issues underlying the type of experiment they devised.

Now why did they propose their analysis at all? They were attempting to study a higher dimensional example of the Bell experiment, to see how the inequality might fare. Before they were deterred and sidetracked from this concern by their errors of logic and of casual notation, they were trying to formalise the implications of the principle of local realism for a four-dimensional experiment. As it turns out, the formalisation can be constructed without confronting any contradiction at all, but we must be a little bit careful. A preliminary airing of a few issues will conclude my analysis here.

## 3. Expanding the CHSH/Bell Formulation to Four Dimensions

Remember that Einstein’s principle of local realism pertains to situations that lie outside the scope of quantum theory. In the context of a GHSZ 4-ply experiment on two pairs of electrons, his assertion concerns the outcome of other experiments that one might consider conducting in tandem with the 4-ply experiment that one does conduct. However, these others are precluded from concomitant execution by the one that is conducted. According to the general form of the uncertainty principle, quantum theory expressly renounces any scope for making proclamations about the joint result of such incompatible ventures. It can and does make probabilistic statements separately about the results of any one of the 4-ply experiments that may be considered. No worries. We can at least think about the prospect of such an impossible joint venture and assess the relevance of what quantum theory does say regarding its conduct. Many, many people have. Well, what would we need to think about if we were to think about such vagaries in the context of the GHSZ experiment? At least we now have a vocabulary and a syntax of language to talk about what we are thinking.

Suppose, for example, that the outcome of a specific run of the experiment characterised by a 4-angles setting of $\left(\phi_{1},\phi_{2},\phi_{3},\phi_{4}\right)$ for which $\kappa =\phi_{1}+\phi_{2}-\phi_{3}-\phi_{4}=0$, yields the observed spin results
$$\left[A_{\left(\phi_{1},\phi_{2},\phi_{3},\phi_{4}\right)}\left(\phi_{1}\right),B_{\left(\phi_{1},\phi_{2},\phi_{3},\phi_{4}\right)}\left(\phi_{2}\right),C_{\left(\phi_{1},\phi_{2},\phi_{3},\phi_{4}\right)}\left(\phi_{3}\right),D_{\left(\phi_{1},\phi_{2},\phi_{3},\phi_{4}\right)}\left(\phi_{4}\right)\right]\;=\;\;\left(+1,-1,-1,-1\right).$$

Notice just to begin that the spin-product of the components of this supposed result vector is −1, as is required for a setup specifying this value of κ=0. Now about this observed result vector, the principle of local realism would say, among other things, that if we were to conduct concomitantly a companion experiment *on this same quartet of electrons* at slightly adjusted angle settings of the form $\left(\phi_{1},\phi_{2}+t,\phi_{3},\phi_{4}+t\right)$, for an example that preserves the value of κ=0, then the resulting observation vector for this alternative experiment would only have to satisfy the form
$$\left[\;A_{\left(\phi_{1},\phi_{2}+t,\phi_{3},\phi_{4}+t\right)}\left(\phi_{1}\right),\;B_{\left(\phi_{1},\phi_{2}+t,\phi_{3},\phi_{4}+t\right)}\left(\phi_{2}+t\right),\;C_{\left(\phi_{1},\phi_{2}+t,\phi_{3},\phi_{4}+t\right)}\left(\phi_{3}\right),\;\;D_{\left(\phi_{1},\phi_{2}+t,\phi_{3},\phi_{4}+t\right)}\left(\phi_{4}+t\right)\;\right]\;=\;[\;+1,\;x,\;-1,\;x\;],$$
where the value of *x* might equal either −1 or +1. Either value would ensure that the product of the component observations equals −1 as required. The claim of local realism is that the behaviour of these electrons at stations 1 and 3 would remain the same in any other run in any other setting of the angles at stations 2 and 4, and moreover that the prescriptions of quantum probabilities must be preserved. If the twists of the magnets at 2 and 4 would destroy the value of κ in the concomitant run, then the spin values at 1 and 3 *for this same pair of electrons* would still remain the same according to local realism, but the concomitant spin values at 2 and 4 might enjoy more liberty. This would depend on the value of κ associated with the adjusted magnet angles $\phi_{2}^{\prime }$ and $\phi_{4}^{\prime }$. If the value of κ for the adjusted magnet angles were not equal to either 0 or π then the spin values at stations 2 and 4 would be permitted to alternate in sign. This would be allowed because the joint distribution of the component observations (A,B,C,D) would be different in the new angle settings according to quantum theory, and even the QM expectation of the of the product ABCD would then be different. It would be only the spin values at angles unchanged that would be forced to remain the same in the two concomitant runs on the same pairs of photons.

It is because the electrons are entangled that the specification of the consequences of local realism must take all four angle settings into account in the higher dimensional context. Both of these distinct spin vector observations (1,1,−1,1) and (1,−1,−1,−1) would ensure that the product of the spin values would equal −1 when the angles $\phi_{2}$ and $\phi_{4}$ are both twisted by the same incremental value *t*. So there are two possible observation vectors for this concomitant experiment that would satisfy this proclamation of local realism, but there are many more impossibilities that would not. For examples, an observation vector (1,1,−1,−1) would be impossible in the twisted case, since it does not even respect the tenets of quantum theory. (Remember we are presuming that $\kappa =\phi_{1}+\phi_{2}+\phi_{3}-\phi_{4}=0$, so the product of the four observations must equal −1.) Further, an observation of (1,−1,1,1) would be impossible as well, not being in accord with local realism. Since the value of $C_{\left(\phi_{1},\phi_{2},\phi_{3},\phi_{4}\right)}\left(\phi_{3}\right)$ is proposed as −1 in the contextual angle setting initially proposed, local realism would require it to equal −1 in the second setting as well for which $\phi_{3}$ is the same. This second disallowed result would not accord with the principle of local realism, even though the proposed component product equals +1. Let us think some more.

The principle of local realism specifies that while the results of a quantum experiment may well be stochastic, assigned various probabilities according to the tenets of quantum theory, if the results at the observation stations 1 and 3 in any actual experimental instance equal +1 and −1, say, then these results would have had to be the same in their instantiation in other circumstances of their companion angles as well. If these two electrons had been arriving at their stations in tandem with the other two electrons arriving at their observation stations at different angles than they had, then the results at these stations 1 and 3 would need to remain the same.

I shall now outline a construction of a Bell quantity in CHSH form that would be appropriate to the four dimensional problem of GHSZ. I believe I have delineated the relevant details allowing how it might be done. As befits the pleasures of quantum physical theorising, there will be intrigue involved. What I plan to do is to design a GHSZ gedankenexperiment on a single quartet of doubly paired electrons under 16 distinct conditions. I shall use alternate magnet orientations at the opposing stations 1 and 3 and at the opposing stations 2 and 4 that mimic the four polarisation settings used by Aspect in his experiments with single pairs of photons. You should recognise that experiments at stations 1 and 3 on their own (ignoring the parallel experiments at stations 2 and 4) would replicate the simpler experiments of Aspect/Bell at stations *A* and *B*.

### 3.1. Setting the Context

We shall now embark on details that require us to clarify two peculiarities of the GHSZ depiction of their 4-ply experiment, which we have reprinted as our Figure 1. To begin, examine the two (x,y) axis portrayals that are situated perpendicular to the $z^{+}$ and $z^{-}$ axes at which a pair of electrons would enter stations 1 and 3. On their own (ignoring for now the parallel axis systems relevant to stations 2 and 4) they constitute a replication of the planes in which the angling pair of polarisers were situated in the Aspect/Bell problem. In the GHSZ situation, the electron passing the S-G magnet at station 1 enters the complex from the source of its propagation, and the electron passing the magnet in the (x,y) plane at station 3 enters the complex from the same source. Because this propagation source is between the two station planes, the electrons are travelling in the opposite directions, these being $z^{+}$ and $z^{-}$. As the two (x,y) planes they enter are designed to be identical, the plane depicted at station 3 in the GHSZ Figure is displayed incorrectly, as a reflection (about the vertical x-axis) of its true orientation. Visualise the directional vector depicted at station 3 but with the y-axis rotated by π radians about the x-axis. Think about it as you view the axis from behind, entering from the propagation source. The depicted vector sweeps an angle of π+ϕ/3 from the x-axis rather than $\phi_{3}$ as labelled. You will get it.

Fair enough, it is only a picture, and we can easily adjust our understanding of the situation for this miscue. For the algebraic derivation of the expectation formula $E^{\Psi }\left[ABCD\left(\phi_{1},\phi_{2},\phi_{3},\phi_{4}\right)\right]$ in their Appendix F seems to be appropriate to the real orientations of the (x,y) planes at each station. In the orientations displayed, the GHSZ Figure has the benefit that it suggests correctly that the (x,y) axis entered by each electron as it passes the magnet appears identical. It would only be the direction of the S-G magnets that might vary from station to station.

A second peculiarity of the displayed Figure merits mention. Notice that at all stations the (x,y) axis is depicted with the x-axis being vertical and the y-axis being horizontal. This is unusual relative to the common arbitrary orientation of these axes in which the x-axis runs horizontally from left to right as the value of *x* increases, while the y-axis runs vertically from bottom to top. There is nothing improper about this if the situation is recognised for what it is. The upshot of their non-standard convention relative to the sizes of the angles $\phi_{1},\phi_{2},\phi_{3},\phi_{4}$ can be understood by considering the angle pairings specified by the directional vectors $a,b,a^{\prime }$, and $b^{\prime }$ in the original Aspect experiment. Using the standard convention of the directions of the (x,y) plane, it is clear that the directional angles of Aspect’s polarisers would be 7π/16,5π/16,3π/16, and π/16. These would make the four angles Aspect denoted by $\left(a,b\right),\left(a,b^{\prime }\right)$, and $\left(a^{\prime },b\right),\left(a^{\prime },b^{\prime }\right)$ equal to −π/8,−3π/8,π/8, and −π/8 which I have used in my analysis of the Aspect/Bell conundrum [1]. (Remember that the angle denoted by (a,b) for example, is the angle passing between the directional vectors a and b.) I think we can be clear in proceeding now with some numerical investigations.

### 3.2. Now Pursuing the Gedankensetup

I shall now provide an analysis of sixteen simultaneous gedankenexperiments at which the two pairs of electrons are sent to the four stations, concomitantly toward two different magnet angles at each station. I shall be brief, merely reporting the results of the analysis at each stage of its development rather than providing every interim detail.

The two chosen angles at each station *i* will be designated as $\phi_{i}$ and $\phi_{i}^{\prime }$. According to the principle of local realism, we presume that any specific instantiation of a spin observation at a station at a specific angle would arise as the same, no matter which other three angles were engaged by the electrons at the other stations. This is despite the fact that the QM joint probability distribution for the outcome of all four observations (and for their product) clearly depends on the settings of all four magnet angles. This is exactly the principle we followed in generating results of the simpler gedankenexperiment on single photon pair in the framework of Aspect/Bell. Identifying the ensemble of possible observations (the realm matrix) for the eight spin results exhibited in the sixteen experiments involving eight arbitrary angles, we would describe it in transposed form as $R{\left[A\left(\phi_{1}\right),B\left(\phi_{2}\right),C\left(\phi_{3}\right),D\left(\phi_{4}\right),A\left(\phi_{1}^{\prime }\right),B\left(\phi_{2}^{\prime }\right),C\left(\phi_{3}^{\prime }\right),D\left(\phi_{4}^{\prime }\right)\right]}^{T}$ and would recognise the matrix columns as composing the Cartesian product of eight component vectors, ${\left\{-1,+1\right\}}^{8}$.

Now beneath each column of this 8×256 matrix, we would exhibit the vector of 4-product spin results that corresponds to this possibility at each of the sixteen scenarios of the concomitant gedankenexperiments. Sequentially, this vector would designate the ordered 4-spin-products using none, one, two, three, and then four of the “alternate” primed magnet angles $\phi_{i}^{\prime }$ rather than the base orientation at angle $\phi_{i}$. Specifically, this section of the full realm matrix would be designated as follows, with the values of κ and of the QM-motivated expectation values relevant to each component product shown to its right.
$$\begin{array}{ccccc} R[A\left(\phi_{1}\right)B\left(\phi_{2}\right)C\left(\phi_{3}\right)D\left(\phi_{4}\right) & & \pi /4 & & -1/\sqrt{2}\\ A\left(\phi_{1}^{\prime }\right)B\left(\phi_{2}\right)C\left(\phi_{3}\right)D\left(\phi_{4}\right) & & 0 & & -1\\ A\left(\phi_{1}\right)B\left(\phi_{2}^{\prime }\right)C\left(\phi_{3}\right)D\left(\phi_{4}\right) & & 0 & & -1\\ A\left(\phi_{1}\right)B\left(\phi_{2}\right)C\left(\phi_{3}^{\prime }\right)D\left(\phi_{4}\right) & & \pi /2 & & 0\\ A\left(\phi_{1}\right)B\left(\phi_{2}\right)C\left(\phi_{3}\right)D\left(\phi_{4}^{\prime }\right) & & \pi /2 & & 0\\ A\left(\phi_{1}^{\prime }\right)B\left(\phi_{2}^{\prime }\right)C\left(\phi_{3}\right)D\left(\phi_{4}\right) & & -\pi /4 & & -1/\sqrt{2}\\ A\left(\phi_{1}^{\prime }\right)B\left(\phi_{2}\right)C\left(\phi_{3}^{\prime }\right)D\left(\phi_{4}\right) & & \pi /4 & & -1/\sqrt{2}\\ A\left(\phi_{1}^{\prime }\right)B\left(\phi_{2}\right)C\left(\phi_{3}\right)D\left(\phi_{4}^{\prime }\right) & & \pi /4 & & -1/\sqrt{2}\\ A\left(\phi_{1}\right)B\left(\phi_{2}^{\prime }\right)C\left(\phi_{3}^{\prime }\right)D\left(\phi_{4}\right) & ,\mathrm{for}\;\mathrm{which}\;\;\;\;\kappa \;\;=\;\; & \left(\;\pi /4\;\right) & \;\;\;\;\;\;\;\;\;\mathrm{and}\;\;\;\;\;E\;\;=\;\; & \left(\;-1/\sqrt{2}\;\right)\\ A\left(\phi_{1}\right)B\left(\phi_{2}^{\prime }\right)C\left(\phi_{3}\right)D\left(\phi_{4}^{\prime }\right) & & \pi /4 & & -1/\sqrt{2}\\ A\left(\phi_{1}\right)B\left(\phi_{2}\right)C\left(\phi_{3}^{\prime }\right)D\left(\phi_{4}^{\prime }\right) & & 3\pi /4 & & +1/\sqrt{2}\\ A\left(\phi_{1}^{\prime }\right)B\left(\phi_{2}^{\prime }\right)C\left(\phi_{3}^{\prime }\right)D\left(\phi_{4}\right) & & 0 & & -1\\ A\left(\phi_{1}^{\prime }\right)B\left(\phi_{2}^{\prime }\right)C\left(\phi_{3}\right)D\left(\phi_{4}^{\prime }\right) & & 0 & & -1\\ A\left(\phi_{1}^{\prime }\right)B\left(\phi_{2}\right)C\left(\phi_{3}^{\prime }\right)D\left(\phi_{4}^{\prime }\right) & & \pi /2 & & 0\\ A\left(\phi_{1}\right)B\left(\phi_{2}^{\prime }\right)C\left(\phi_{3}^{\prime }\right)D\left(\phi_{4}^{\prime }\right) & & \pi /2 & & 0\\ A\left(\phi_{1}^{\prime }\right)B\left(\phi_{2}^{\prime }\right)C\left(\phi_{3}^{\prime }\right)D\left(\phi_{4}^{\prime }\right)] & & \pi /4 & & -1/\sqrt{2}. \end{array}$$

GHSZ quite rightly derive the QM-motivated expectations for each one of these 4-spin-products via the specification $E^{\Psi }\left[A\left(\phi_{1}^{*}\right)B\left(\phi_{2}^{*}\right)C\left(\phi_{3}^{*}\right)D\left(\phi_{4}^{*}\right)\right]\;=\;-cos\left(\phi_{1}^{*}+\phi_{2}^{*}-\phi_{3}^{*}-\phi_{4}^{*}\right)$, where each angle designation $\phi_{i}^{*}$ might represent either $\phi_{i}$ or $\phi_{i}^{\prime }$.

Having studied the simpler problem of Aspect/Bell, it is now no surprise to find that the realm matrix so generated for the sixteen 4-spin-products does not display 256 distinct columns, but rather, merely 32. What this tells us is that if we can identify any five rows of this 16×256 matrix whose columns exhaust the Cartesian product ${\left\{-1,+1\right\}}^{5}$, then the remaining rows will have been identified to be functionally related to them. It turns out that there are many such combinations of five rows that will do this. Of the ${}^{16}C_{5}=4368$ available for such choices of five rows, 2688 of them provide a functional relation. In fact, all of these functional relations hold among the sixteen components of any column vector in the realm. Although this is quite a step up in the complexity of the functional restrictions among the 4-spin-products of a gedankenexperiment, it is structurally no different than the four simultaneous functional relations that bound up the 2-polarisation-products in the simpler context of the Aspect/Bell problem.

Let us stop for a moment to think about what this means: the conditional distribution for the other eleven 4-spin-products given any of these five is degenerate on the function values they stipulate. Ostensibly, we are searching for a QM-motivated joint probability distribution for the results of all sixteen 4-spin-product quantities in the GHSZ gedankenexperimental design. Of course the general uncertainty principle tells us that there is no such joint distribution, because quantum theory says nothing about the joint result of any two such incommensurable 4-products. However, if there were such a distribution then it could be factored as the product of the marginal joint distribution for any five of them times the conditional distribution for the other eleven given these five. Since this conditional distribution relative to any of the 2688 function-producing choice of five from sixteen is degenerate, the joint distribution for all sixteen would resolve to a specification of the joint distribution for these five. However quantum theory does not uniquely identify such a distribution. What it does do however, is specify the expectations of the products of any four spin values that might be observed in an experiment in any 4-angle setting. Along with the summation constraint, the five such product expectation values places six linear constraints on the probabilities for the thirty-two constituents of the partition underlying the joint distribution. It is precisely this situation to which the computational structure of linear programming problems applies.

Well what should be the objective function of such a linear programming problem? We have seen that the Bell quantity “*s*” in the Aspect/Bell analysis of the CHSH formulation is a linear combination of the Expected products of all four components of the gedankenexperiment involved. The coefficients identifying the linear combination are all either +1 or −1. Therefore, consider as the objective function of the linear programming problem we might formulate for the double electron pair experiment as a similar linear combination of the sixteen 4-spin-products we are considering. Five linear constraints on the spin-products for the function domain settings plus the summation constraint would leave 26 dimensions of freedom in the argument vector of the objective function. The linear programming computation would find the vectors $q_{32}^{min}$ and $q_{32}^{max}$ that yield the extreme values of this linear combination.

To compute a complete solution to the gedanken problem we would need to compute 2688 pairs of LP problems (*min* and *max*) to find all the vertices of the polytope of probability distributions that quantum theory would allow as its solution. Of course, as in the corresponding Aspect/Bell solution, many of these would be duplicates. At any rate, we would arrive at a larger sized solution of the quantum theory specified polytope of probability distributions that is structurally similar to the solution we found in the case of Aspect/Bell. End of story.

The only concluding remark is that every one of the feasible expectations for the objective function would respect the bounds specified by any appropriate Bell-type inequality.

## 4. Technical Conclusions

Quite to the contrary of the GHSZ conclusion that the premises of EPR pose a contradiction to a quantum experiment involving four (and even only three) particles, we can conclude that such experiments indeed do allow the premises of EPR, appropriately identified. Furthermore, the conditions of such an experiment can easily exemplify well the EPR premise of “perfect correlation,” if desired, in both the case of conditions of (12a) and that of (12b)—though not at the same time! Smile.

## 5. Concluding Comments

The results of this discussion have laid bare the claims of GHSZ who denigrate the logical consistency of the EPR version of local realism. However, I would not like a reader to think that I am in any way an advocate of the EPR construal of the situation either. From a larger perspective, I find it embedded in a view of physical experience that is seriously out of date. My investigations of the past five years have been oriented quite narrowly, to a resolution of the conundrum posed by the purported violation of Bell’s inequality in the theory of quantum mechanics in any of the forms in which it has been promoted. At a deeper level, I conclude that the so-called mysterious properties of the quantum world, involving a structure of “quantum probabilities” which inhere different structures than the mundane probabilities of the world at the classical scale, have been misconstrued as well. A discussion of larger implications relevant to a reconstruction of physical theory awaits a confirmation of my narrow concerns.

As to the definitive empirical work of Hensen et al. [12], I do not doubt their experimental results. However, their statistical analysis suffers from the same mistake as does that of Aspect. Readers still impressed by it might review Section 6 my manuscript [1] on the mistaken violation of Bell’s inequality. Nonetheless, I do sorely doubt the conclusion of Wiseman [13] proclaiming “death by experiment to local realism”.

## Figures and Tables

**Figure 1 entropy-22-00759-f001:**
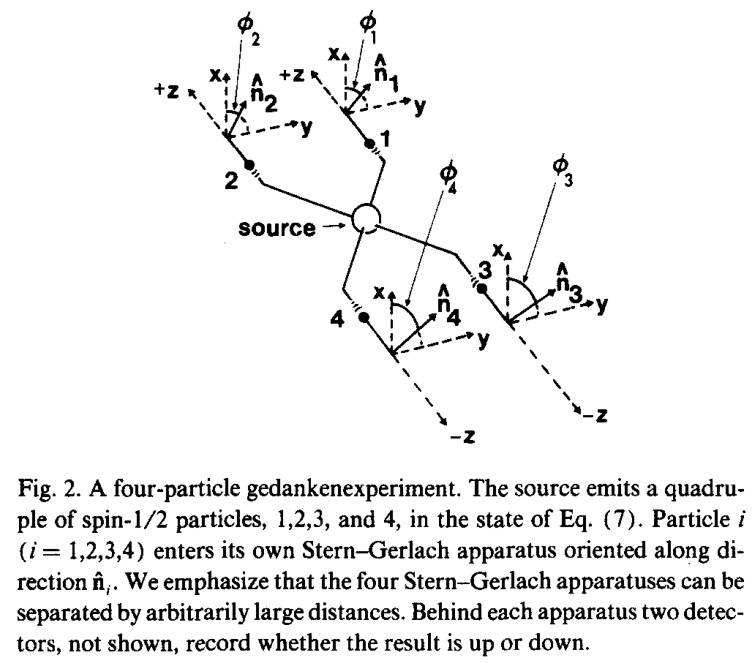
The original schema of the GHSZ article, reproduced with permission from *American Journal of Physics*, **58**(12), page 1134, doi:10.1119/1.16243. https://aapt.scitation.org/doi/10.1119/1.16243#.

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
