# Peer review of "The GHSZ Argument: A Gedankenexperiment Requiring More Denken"

_entropy, 2020, doi:10.3390/e22070759_

Round 1

Reviewer 1 Report

This is an excellent paper that clearly describes the logical-mathematical deficiencies of the work of GHSZ. It does unfortunately not reference some important related work. Aschwanden et al. (probably with less generality) had already presented a counterexample to the GHZ work as well as the experiments by Pan et al.

Reference and a short description of its relation should be given to the work of Manuel Aschwanden et al "Local Time Dependent Instruction-Set Model for the Experiment of Pan et al.",  which was published in Quantum Theory, Reconsiderations of Foundations 3, AIP Conference Proceedings 810, G. Adenier, A. Y. Khrennikov and T. M. Nieuwenhuizen Editors, Melville, New York, 2006.

A description of this work would fit at the place where Lad explains the impossibility of the simultaneous occurrence of certain events.

Author Response

For Reviewer #1

Thank you for your comments.  I was very pleased that you recognised the importance of my results and the analysis I provided.  I think this will be a notable and controversial paper.  I did study both the paper of Aschwanden et al which you suggested, and the original of Pan et al which they parleyed, and I found their empirical assessments quite relevant.    I inserted a paragraph to this effect exactly in the place where you suggested it. Thanks. 

However I appended to this statement another comment on the empirical work of Pan et al, but only as a footnote.  As you are familiar with the claims of the Pan group regarding three polarizations in the GHZ state, I hope you will be pleased to think about what I have written there.  It is written tersely, written to be understandable to readers who are well familiar with the argument of Pan et al, which is sorely mistaken on even more fundamental grounds.  Please see what you think of that footnote.  Because my analysis of the GHSZ paper is so extensive, I did not think this was the place to get into a similarly exhaustive reconstruction of the three-photon problem, but I hope you will get the idea of my view of what is wrong with it fundamentally by thinking about the footnote.

I am not sure if you have seen my companion paper on the Violation of Bell's inequaltiy: a misunderstanding ...  which I have submitted to this same special issue of Entropy.  I am proposing that it should be read prior to this one on GHSZ.  If you have not seen this manuscript then you can find a version of it on my Researchgate site.  At any rate, if you do look into my Researchgate site you will find still another challenging paper which may interest you:  Hoojums more than Boojums, quantum mysteries for no one.  The reason it may interest you is because it addresses the issue of encoded messages Mermin proposes (and relegates) that might identify the content of supplementary variables.   These are related (at least conceptually)  to the instruction sets that Aschwenden et al provide to explain the results of Pan et al.  Anyhow,  thanks again and best wishes.

Reviewer 2 Report

This manuscript reconsiders the argument for "Bell's theorem without inequalities" from Greenberger, Horne, Shimoney and Zeiling, and argues that it results from flawed logic.

It is a well documented phenomenon that quantum theory has long been giving headaches to logician, physicist and virtually most well-constituted human considering its claims of interference, tunneling, contextuality, Bell nonlocality etc. Nonlocal games, such as the GHZ game (by some of the authors of the manuscript discussed here), are precisely buit on paradoxes, or the violation of "conditions of possible experience" as G. Boole himself called them.

Yet, if nature is to have the last word, experiments demonstrate such phenomenons with a confidence rarely achieved anywhere else. In particular, the number of hypotheses needed to test a Bell inequality is so small that it gave rise to a whole new approach to science by itself, so-called device-independent. A vibrant community of physicists and computer scientist mostly has been studying these topics for decades now. This approach has been so fruitful that the space here would be too short to list all relevant papers discussing these topics.

On the point of experimental demonstration, it seems also important to highlight that although the experimental paper cited here by Hensen et al., is criticized for its limited statistical evidence, other works performed simultaneously do not suffer from this drawback, such as Giustina et al., Physical Review Letters. 115 (25): 250401 (2015), Shalm et al., Physical Review Letters. 115 (25): 250402 (2015), Rosenfled et al., Phys. Rev. Lett. 119, 010402 (2017). The last work, for instance, provides enough statistical evidence to reject hidden variables with a p-value of ~3e-9. Before diving into technical details, it seem meaningful to question verify whether a consequence of the current work would be known to be already incompatible with existing evidence: can these experimental Bell violation be reconciled with the argument presented here?

In general, it seems important to keep in mind that physicists have also long been confused by the quantum predictions. Therefore, contrary to other disciplines, it is not always helpful to come back to earliest works, as these might include confusions (and possibly flawed logic) which can now be avoided. An example of this is the notion of 'realism', which although sloppy, has long seemed to be central to the argument of Bell inequalities. In this respect, it is helpful to follow the point of view advertised by J. Bell himself of 'local causality' rather than 'local realism'. These two sets of assumptions both imply the validity of Bell inequalities, but the former is more operational, and hence much less subject to interpretation (especially if violated). The recent review Brunner et al., Rev. Mod. Phys. 86, 419 (2014) could be a good starting point on this.

In my opinion, a clear discussion of how the proposed argument is compatible with existing experimental evidence should be provided. As it stands, it seems that natural evidence invalidates the argument.

Author Response

To Reviewer #2:

I appreciate your consideration of my manuscript, thank you, but I cannot agree with your evaluation.  I am very pleased that the other two reviewers have recognised its consequence and that the editor has determined to publish it with revisions involving citations of relevant literature.

#### I am aware, and I agree that literature on the Bell inequality violations is indeed vast, and I cannot hope to present a literature review and also present my analytical assessment of the GHSZ argument with the detail that I do.  Indeed, several such extensive reviews have been published.  I have made effort in this revised submission to account for this, including a reference to the literature review of Brunner et al which you suggested, as well as references to the substantive text of Greenstein and Zajnic suggested by the editor, and the encyclopedic assessment in the text of Jaeger.  All three of these laud the work of GHSZ which I criticize, and they are both replete with relevant literature references. 

#### As to your suggestion that extensive statistical evidence has been presented that substantiates the violation of the inequality, I can only beg to differ.  I agree that empirical assessments have been disseminated, but I do not agree with their relevance or validity.  I am not sure whether you have looked at the companion submission that I have made to this special issue of Entropy, entitled Violation of Bell's inequality: a misconception based on a mathematical error of neglect. (It is accessible on my Researchgate page if I have not seen it and if you might be interested by it.) In that manuscript I have identified an error that runs through all the many published empirical papers that support what is by now ubiquitously touted as ``overwhelming evidence in favour of quantum theoretical probabilities that are said to defy the principle of local realism''.  Moreover, I show clearly that the claim that the expectation of the CHSH quantity "s" equals 2\sqrt{2} (approximately 2.82, outside of the Bell interval [-2,+2] ) is mistaken.  In addition, the error underlying Aspect's supposed empirical support of this claim is corrected in that manuscript.  It is shown that quantum theory does not provide support for the 2\sqrt{2} proposal.  Rather, quantum theory, guided by the uncertainty principle (agnostic regarding Hadamard operators that do not commute) actually supports not any specific number at all, but rather only an interval specified in rounded form as (1.1213, 2).  In that manuscript I have pointed out the error involved in the empirical evaluation of Bell's inequality in the style begun by Aspect which has been continued to be followed in the evaluation of evidence from subsequent experimentation.  A motivational computation simulating Aspect's results supports a measure of 1.7667, and it is shown why this cannot be definitive in preference to the interval.  All the p-values in the world based on mistaken computations cannot relieve this conclusion.  It is a commentary on our times that repetitions of fallacious statements are considered demonstrable evidence for their truth.

It is true that empirical research in physics has progressed extensively since the seminal papers that have begun this train of research.  However, this is not a reason to neglect early seminal papers, but rather an impetus to identify precisely fundamental and influential errors in the pathbreaking papers when they have led followers astray.  I do not mean to be pompous, but this is what I have done in highlighting my reconstructive assessments of both the Aspect/Bell results and those of GHSZ.  I believe they deserve a hearing and a professional discussion.  I also believe that John Bell would have been very interested to hear of my insights, as he had long suspected there was something wrong with the supposed defiance of his inequality by quantum probabilities.

Thanks again for your attention to my paper.  I hope we can engage in collegial discussion at some time in the future.   Sincerely,  Frank Lad

Reviewer 3 Report

This paper presents a thorough, insightful, and original analysis of the GHSZ thought experiment. The arguments are very clear and supported by what seems to be the first logically correct treatment of the problem. The paper is very well written.

This paper deserves to be published as it is.

The following comments are merely to share with the author.

  • The subscript lambda for A, B, C, and D GHSZ used is the hidden variable. The subscript the author used refers to the settings of the experiment, which is typically used for data analysis. Conceptually, they are different. But this does not change the nature of the logical mistake GHSZ made. The analysis of logical mistake could be done in terms of hidden variable too.
  • To my opinion, the references should/could be improved a little bit. The author should try (or has tried) to search for the existing literatures to see if there already exist similar arguments or conclusions in the Bell stuffs. For example, some authors pointed out that the quantities of all the correlations in the Bell’s type of settings cannot be all independent with each other.
  • I would point out a paper written by Boole published before QM came, which might be interesting to the author. The Bell type inequalities are presented there.
  1. Boole, “On the theory of probabilities”, Philos. Trans. R. Soc. London 152, 225 (1862)

Author Response

For Reviewer #3:

Thank you very much for your assessment of my submitted manuscript.  I think it will be a notable and controversial paper.  I do hope that it will be printed following the first in a series of three that I have submitted, but I am not sure of that.  I am also not sure whether you have also seen that manuscript, my assessment of Violation of Bell's inequality: a misunderstanding ...   which can be found on my Researchgate page.  That is where the functional relations among the four components of the CHSH quantiy "s" are identified and accounted for.  I wish that I could find some previously published allusions to dependencies among the components of the linear combination defining the CHSH quantity "s" (the subject of Bell's inequality in that form) that you suggest, for my references.  But I have not found any.  I have been paying attention to ArXiv manuscripts relevant to Bell for some years now, but I know of nothing appropriate.  If you could provide me with a title I would be pleased, as you have written "For example, some authors pointed out that the quantities of all the correlations in the Bell’s type of settings cannot be all independent with each other."  I don't know of these.  I can say that ArXiv refused to disseminate an earlier version of my paper on the error of Aspect/Bell a few years ago (which provides analytic detail of this) on the strange grounds that they did not have the expertise to review it. ???  So they were protecting the established result.

As to the hidden variables lambda, I hope I have made clear that I have eliminated it in my text only because it appears as a subscript to every (!) instance of a spin quantity A,B,C,D in the GHSZ manuscript and thus performs no practical function there.  I just wanted to use that position in the notation then for the useful contextual purpose that is required.  Of course the universal content of lambda, which represents the possible supplementary variables, is crucial to the whole story, the very reason that GHSZ get involved in their analysis to begin with.  My third submission addresses the formalities of characterising the supplementary variables position, for which there is a now sizeable literature on supposed "impossibility theorems" stemming from the original claims of von Neumann.  It is also up on the Researchgate page.

As to Boole, I have also been taken by his works.  You might enjoy too my commentary on his relation to the technical analysis of Bruno de Finetti in my book   "Operational subjective Statistical Methods: a mathematical, philosophical, and historical introduction. 1996, New York: John Wiley.  See pp 26, 54, 89,105, and 136.

Thanks again for your attention to my work.  Best wishes.  Frank Lad

Hello again Reviewer #3, Just to let you know, I have learned of the typ[e of resweaqrch that is in sync with mine, with several interesting references. IThere is no need for more info on that count from either of you. I can assure you that I will have a sentence or two to include in my final revised ms that will place my work in the context of others who have similar reactions to the proposed violations of Bell's inequality. Whatever else you have to say on the matter, I will be able to provide this emendation without need of further reference from you, for which I had asked. Thanks very much. Frank